# The Anti-Inflammatory and the Antinociceptive Effects of Mixed *Agrimonia pilosa* Ledeb. and *Salvia miltiorrhiza* Bunge Extract

**DOI:** 10.3390/plants10061234

**Published:** 2021-06-17

**Authors:** Jing-Hui Feng, Hyun-Yong Kim, Su-Min Sim, Guang-Lei Zuo, Jeon-Sub Jung, Seung-Hwan Hwang, Youn-Gil Kwak, Min-Jung Kim, Jeong-Hun Jo, Sung-Chan Kim, Soon-Sung Lim, Hong-Won Suh

**Affiliations:** 1Department of Pharmacology, College of Medicine, Hallym University, 1 Hallymdaehak-gil, Chuncheon 24252, Gangwon-do, Korea; B17501@hallym.ac.kr (J.-H.F.); sumin@hallym.ac.kr (S.-M.S.); 2Institute of Natural Medicine, College of Medicine, Hallym University, 1 Hallymdaehak-gil, Chuncheon 24252, Gangwon-do, Korea; de3180@hallym.ac.kr; 3Department of Food Science and Nutrition, College of Natural Science, Hallym University, 1 Hallymdaehak-gil, Chuncheon 24252, Gangwon-do, Korea; khy9514@nate.com (H.-Y.K.); B16504@hallym.ac.kr (G.-L.Z.); isohsh@gmail.com (S.-H.H.); 4R&D Center, Huons Co., Ltd., 55 Hanyangdaehak-ro, Ansan 15588, Gyeonggi-do, Korea; 5Research Institute, Huons Nature, Geumsan 32742, Choong-cheong Nam-do, Korea; kyg@huonsnature.com (Y.-G.K.); jas.mjkim@gmail.com (M.-J.K.); jhjo@huonsnature.com (J.-H.J.); 6Department of Biochemistry, College of Medicine, Hallym University, 1 Hallymdaehak-gil, Chuncheon 24252, Gangwon-do, Korea; biokim@hallym.ac.kr

**Keywords:** *Agrimonia pilosa* Ledeb., *Salvia miltiorrhiza* Bunge, *Perna canaliculus*, mixture extract, antinociception, anti-inflammation, osteoarthritis

## Abstract

Arthritis is a common condition that causes pain and inflammation in a joint. Previously, we reported that the mixture extract (ME) from *Agrimonia pilosa* Ledeb. (AP) and *Salvia miltiorrhiza* Bunge (SM) could ameliorate gout arthritis. In the present study, we aimed to investigate the potential anti-inflammatory and antinociceptive effects of ME and characterize the mechanism. We compared the anti-inflammatory and antinociceptive effects of a positive control, *Perna canaliculus* powder (PC). The results showed that one-off and one-week treatment of ME reduced the pain threshold in a dose-dependent manner (from 10 to 100 mg/kg) in the mono-iodoacetate (MIA)-induced osteoarthritis (OA) model. ME also reduced the plasma TNF-α, IL-6, and CRP levels. In LPS-stimulated RAW 264.7 cells, ME inhibited the release of NO, PGE_2_, LTB_4_, and IL-6, increased the phosphorylation of PPAR-γ protein, and downregulated TNF-α and MAPKs proteins expression in a concentration-dependent (from 1 to 100 µg/mL) manner. Furthermore, ME ameliorated the progression of ear edema in mice. In most of the experiments, ME-induced effects were almost equal to, or were higher than, PC-induced effects. Conclusions: The data presented here suggest that ME shows anti-inflammatory and antinociceptive activities, indicating ME may be a potential therapeutic for arthritis treatment.

## 1. Introduction

Osteoarthritis (OA) is one of the most common joint diseases, and is a painful and disabling disease that affects millions of patients [1]. Several factors, including mechanical influences, the effects of aging on cartilage matrix composition and structure, and genetic factors, lead to the increasing prevalence of OA [2]. In the clinical context, OA is being treated with nonsteroidal anti-inflammatory drugs (NSAIDs) or opioid analgesics. However, severe side effects are associated with the long-term use of these drugs, such as peptic ulcers, platelet dysfunction, nausea, constipation, and dizziness [3]. There is a crucial need for specific drugs that can be used in anti-inflammation and analgesia as potential therapies for OA.

*Agrimonia pilosa* Ledeb. (AP), belonging to the Rosaceae family, is distributed over eastern Asia and eastern Europe. AP has been used as a traditional medicinal herb to treat abdominal pain, sore throat, headaches, and parasitic infections in some Asian countries [4]. Further reports showed that AP has antioxidant [5], antinociceptive [6,7], anti-inflammatory [8], and anti-allergic [9] effects. Additionally, *Salvia miltiorrhiza* Bunge (SM), one member of the mint family, is widely used in China, Korea, and Japan [10]. The roots of SM have been reportedly used in traditional treatments of cardiovascular disease and hepatitis [11]. Moreover, SM extract alleviates dysmenorrhea and painful osteoarthritis, which might result from the prompting of blood flow and removal of blood [10,12]. The anti-osteoporotic effect of SM EtOH extract has also been verified recently [10]. According to the pharmacological properties of AP and SM, the combined therapy of AP and SM is likely to attenuate antinociception and protect joins. Furthermore, our previous study demonstrated that the combined therapy of extracts of AP and the extracts of SM shows an antinociceptive effect in the monosodium urate (MSU)-induced gout pain model [13]. Hence, we hypothesize that the mixture extracts (ME) from AP and SM would produce an antinociceptive effect on the mono-iodoacetate (MIA)-induced OA model.

Arthritis can trigger an inflammatory process that involves the activation of immune cells, microglia, and astrocytes. A number of inflammatory cytokines, including C-reactive protein (CRP), leukotriene B4 (LTB_4_), prostaglandin E2 (PGE_2_), interleukin 6 (IL-6), and tumor necrosis factor alpha (TNF-α), play crucial roles in oxidative stress, inflammatory processes, and pathophysiology of OA [14,15,16,17]. The inhibition of such inflammatory cytokines can be an effective approach to decrease articular cartilage loss in arthritis [18]. Both AP and SM have been known to attenuate the release of inflammatory cytokines in lipopolysaccharide (LPS)-induced Raw 264.7 cells [8,19,20,21]. Additionally, AP extracts could inhibit mitogen-activated protein kinase (MAPK)/c-Jun N-terminal kinase (JNK)-mediated inflammation in LPS-activated RAW 264.7 macrophages [22], while promoting the expression of peroxisome proliferator-activated receptor-gamma (PPAR-γ) [23]. Thus, in the present study, we aimed to characterize the mechanism of ME involved in inflammation and arthritis pain in vivo and in vitro.

In recent decades, several studies have reported on the extracts from *Perna canaliculus* (PC) as a compelling candidate for the treatment of arthritis due to their anti-inflammation effect [24,25]. As such, we planned to compare the effects of ME and PC on the animal arthritis and inflammation models. According to previous studies, AP showed antinociceptive effect and SM displayed an anti-inflammatory property. Take it together, our study was designed to prove the achievable therapeutic effect of the extracts of the mixture of AP and SM for the treatment of arthritis. Meanwhile, we planned to investigate the possible mechanism of the effect of mixture extracts on the improvement of arthritis.

## 2. Results

### 2.1. Chromatographic Analysis of ME

High-performance liquid chromatography (HPLC) patterns of ME are shown in Figure 1. The two components, rutin (retention time (t_R_): 13.23) and salvianolic acid B (t_R_: 21.75), were identified by comparing the chromatograms of the standard compounds at 360 nm and 254 nm wavelengths.

### 2.2. ME and PC Reduced Mechanical Pain Threshold in the MIA-Induced OA Model

At 14 days after the MIA-induced OA model was produced, various doses (10, 50, or 100 mg/kg) of ME and PC were orally administered. As shown in Figure 2, both the single and repeated treatments of ME reduced the pain threshold in a dose-dependent manner in the MIA-induced OA model. The antinociceptive effect began to increase at 30 min after the single ME administration, and the paw withdrawal thresholds remained higher than that of the control level even 2 h after ME administration (Figure 2A). In addition, ME still displayed antinociception in the group that was repeatedly administered for 1 week (Figure 2B).

By comparing the antinociceptive effect between ME and PC, the ME induced effect was more effective than PC-induced effect in relieving the pain in the MIA-induced OA model. Moreover, the onset time of antinociception in the ME-treated group was shorter than in the PC-treated group (Figure 2).

### 2.3. ME and PC Suppressed the Levels of Plasma CRP, TNF-α, and IL-6 in the MIA-Induced OA Model

ME and PC were orally administered at the dose of 100 mg/kg 14 days after the production of the MIA-induced OA model. The plasma cytokines concentration was measured using enzyme-linked immunosorbent assay (ELISA) kits, such as CRP, TNF-α, and IL-6. Concerning the difference between ME and PC, similar trends were noted, and the changes in the cytokine release of both groups were shown in Figure 3. Both ME and PC exhibited a significant reduction in cytokine release at the dose of 100 mg/kg.

### 2.4. ME and PC Ameliorated Ear Edema Induced by Croton Oil

As revealed in Figure 4, the application of 2.5% croton oil caused edema of the mice ears, as manifested by weighing measurement. The oral treatment with ME or PC (50 or 100 mg/kg) dose-dependently reduced the weight of the ear (Figure 4).

### 2.5. ME and PC Decreased the Nitric Oxide (NO), PGE_2_, LTB_4_, and IL-6 Release in RAW 264.7 Cells

Various concentrations (from 1 to 100 µg/mL) of ME or PC 1 h prior to LPS (1 µg/mL) were added to the RAW 264.7 cells. At 24 h after the treatment, the media was collected for measuring the levels of NO, PGE_2_, LTB_4_, and IL-6. As shown in Figure 5, significantly higher expression levels of NO, PGE_2_, LTB_4_, and IL-6, were observed in the LPS-induced macrophages. Then, two-way analysis of variance (ANOVA) revealed that the higher concentrations of ME (100 µg/mL) caused reductions in the NO and IL-6 release (Figure 5A,C), and in Figure 5E,G, we see that pretreatment with ME (10 and 100 µg/mL) reduced the LPS-upregulated production of PGE_2_ and LTB_4_ in a concentration-dependent manner. Additionally, PC caused remarkable inhibition of the NO release at concentrations of 10 and 100 µg/mL. The expression of IL-6 was significantly decreased by 100 µg/mL of PC in the macrophages (Figure 5D), while the level of PGE_2_ was not altered (Figure 5H). Moreover, Figure 5H shows that PC (from 1 to 100 µg/mL) mildly suppressed the release of LTB_4_ induced by LPS. Furthermore, the treatment with ME or PC alone did not affect the NO, PGE_2_, LTB_4_, or IL-6 levels without LPS stimulation.

### 2.6. ME and PC Downregulated the Expression Levels of MAPK and TNF-α Proteins in RAW 264.7 Cells

We further examined whether ME or PC would alter the expressions of TNF-α and some MAPK proteins, such as p-ERK and p-JNK proteins. As shown in Figure 6 and Figure 7, LPS treatment caused the upregulation of p-JNK, p-ERK, and TNF-α expression in the RAW 264.7 cells. In addition, the pretreatment with ME or PC (from 10 to 200 µg/mL) attenuated the LPS-induced increase in levels of p-JNK, p-ERK, and TNF-α (Figure 6 and Figure 7 and Appendix A).

### 2.7. ME and PC Upregulated the Phosphorylated Level of PPAR-γ Protein in RAW 264.7 Cells

As PPAR-γ exerts anti-inflammatory effects in macrophages [26], we subsequently analyzed the p-PPAR-γ expression in cell lysates by performing Western blot analysis. We observed that p-PPAR-γ expression was reduced in LPS-induced macrophages as compared to the control group (Figure 8 and Appendix A). Then, phosphorylated PPAR-γ protein was upregulated to a similar extent in ME- and PC-treated macrophages. However, without LPS stimulation, there were no differences in the expression of PPAR-γ between the control group and ME or PC treatment groups, implicating an LPS-dependent mechanism.

## 3. Discussion

The present study showed that both short-term and long-term treatment with ME reduce the pain threshold in a dose-dependent manner in the MIA-induced OA model. Previously, we examined the antinociceptive effect of the extracts isolated from AP or SM and the mixture extracts of AP (50% EtOH) and SM (80% EtOH) in the MSU-induced gout arthritis model [13]. In the present study, we mixed the dry AP and SM, and then an extract was prepared (50% EtOH). Much evidence suggests that synergistic effects improve therapeutic outcomes and safety when different herbs are combined [27]. In particular, we found in the present study that a mixture of AP and SM administered orally once is effective for producing antinociception. Furthermore, repeated administration with ME for 7 days (once/day) still showed a profound antinociceptive effect, suggesting that no tolerance is developed in the production of antinociception by ME.

Several studies focused on the anti-inflammation of the AP extracts. The methanol extract of AP could suppress the production of NO, the expression of PGE_2_, and the release of inflammatory cytokines, such as IL-1β and IL-6, in RAW 264.7 cells [9,28]. Besides this, accumulating evidence supports the antinociceptive effects of the tanshinones isolated from SM. For example, Federica et al. showed that cryptotanshinone produces an apparent long-lasting peripheral analgesic effect in various nociceptive phenotypes, via hot plate tests and tail-flick tests [29]. In addition, cryptotanshinone displayed antinociception in chronic constriction injury-induced neuropathic pain in rats [30]. Additionally, tanshinone IIA improved diabetic neuropathy [31]. Moreover, tanshinones from SM could revert the chemotherapy-induced neuropathy [32]. Tanshinol also has been reported to attenuate OA cartilage destruction [33]. Rutin, apigenin, tanshinone IIA, and Salvianolic acid B, the major components of the AP and SM extracts [13], were reported to produce anti-inflammatory effects [34,35]. Rutin, one of the main compounds isolated from AP [13,36], has been proven to produce an analgesic effect in the murine hot plate model, whereby the analgesic effect of rutin was established [37]. Furthermore, a previous study by Selvaraj et al. confirmed that rutin exhibits peripheral and central antinociceptive activities [38]. Rutin also displayed a protective effect against rheumatoid arthritis via the suppression of NF-κB protein expression [39]. Recently, SM has been reported to attenuate osteoarthritis-related cartilage degeneration through inhibition of the NF-κB signaling pathway [33,40]. Dried SM root contains several biologically active phytocomponents, of which salvianolic acid B is one of the most abundant active phytocomponents with many beneficial properties. In particular, salvianolic acid B was found to act against various auto-immune disorders, including rheumatoid arthritis and alopecia, owing to its potent immunomodulatory and anti-inflammatory activities [41]. As expected, in the present study, ME-pretreated macrophages displayed an anti-inflammatory phenotype upon LPS stimulation characterized by inhibiting the release of pro-inflammatory cytokines NO, PGE_2_, LTB_4_, and IL-6, and upregulating the anti-inflammatory cytokine p-PPAR-γ. Moreover, the ME treatment of OA mice suppressed the production of pro-inflammatory cytokines IL-6, CRP, and TNF-α in plasma, and increased the pain threshold of the hind paw. These results may shed light on how ME could protect joints from arthritis due to its antinociceptive and anti-inflammatory activity. Thus, ME may be applied as a bioactive composition in functional foods.

Furthermore, we tried to evaluate the antinociceptive and anti-inflammatory properties of ME compared to PC, a well-known anti-arthritis candidate. In the past few decades, numerous studies have been published exploring the role of PC in inflammation and arthritis [24]. Here, in arthritic ICR mice, marked anti-inflammatory changes were observed in the PC-treated group, compared with the control group, which is supported by several other teams. They indicated that PC could reduce inflammation in arthritic animals [42,43] and arthritic humans [24,44]. Various factors were involved in this anti-inflammatory effect, one of which is reducing the release of proinflammatory mediators, such as IL-1, IL-2, IL-6, and TNF-α [42,43,45]. Similarly, following 7 days of MIA stimulation, we observed reductions in CRP, TNF-α, and IL-6 production in PC-treated groups. In our study, the treatment of PC also suppressed NO and IL-6 secretion in the LPS-induced macrophage and downregulated the expression of the TNF-α protein. Moreover, the MAPK pathway has been considered a typical molecular target for developing anti-inflammatory agents [46]. The activation of MAPKs has been reported to upregulate pro-inflammatory mediators [47,48]. Our results partially support these findings in that the treatment of PC inhibits the phosphorylation levels of ERK and JNK. Therefore, these findings suggest that PC may regulate mouse macrophages’ proliferation at least via the ERK/JNK MAPK pathways during inflammation. When we compare the effects of ME and PC, ME displayed an equal or better pharmacological efficacy compared to PC in both the in vitro and in vivo models. Importantly, at the same dose of ME and PC, ME showed an earlier onset time in the production of antinociception than PC in the MIA-induced arthritis model. Thus, we suggest that ME could be considered a substitute for PC, due to it being convenient to obtain and efficient.

We further found that at doses of 50 or 100 mg/kg, both ME and PC produced an anti-inflammatory effect in the croton oil-induced ear edema model. Various studies performed on AP or SM showed that the extracts isolated from AP or SM exert anti-inflammatory effects in ear edema models, which partially supports our findings [22,28,49,50]. For example, the AP ethanol extract at the dose of 2000 mg/kg produced remarkable suppressive effects in the xylene-induced ear edema in mice [22]. Tanshinones isolated from SM (95% ethanol) also significantly inhibited xylene-induced ear edema at a dose of 80 mg/kg [28]. Despite being used in the different models of ear edema, our results indicated that at the same potency as PC, ME actually reduces inflammation-induced edema in vivo.

## 4. Materials and Methods

### 4.1. Materials Preparation

Dried AP (leaf) and SM (root), purchased at a local market in Yeongcheon, Gyeongsangbuk-do Province, Korea (June 2019), were authenticated by Emeritus Prof. H. J. Chi, Seoul National University, as in our previous study [13]. These materials were ground to granule size before extraction. The mixture of powdered AP (80 kg) and SM (20 kg) was extracted in 50% EtOH (1200 L × 2 times) at 70 °C for 7 h and filtered with a 60-mesh filter. The extract was lyophilized and yielded brown powders. The extracts were characterized and purified as described previously [6,13,36,51]. In summary, a simple, accurate, and rapid HPLC has been developed to quantify these two polyphenols in a mixed extract and it was successfully validated [13]. The mixture extract contained 1.49 mg/g of rutin and 28.86 mg/g of salvianolic acid B as a specific ingredient, as assessed by HPLC analysis. Here, rutin hydrate (R0035, Tokyo chemical industry Co., Ltd., Portland, ME, USA) and salvianolic acid B (SML0048, Sigma-Aldrich Inc., Louis, USA) were used as the reference materials (Figure 1). The stabilized PC powder (Pernatec^TM^, 931200) was purchased from Waitaki Biosciences (Christchurch, New Zealand). Reagents and solvents were purchased from Sigma-Aldrich Co (Louis, MO, USA).

### 4.2. Phytochemical Analysis Using HPLC

Chromatographic analysis was performed using an HPLC system (Agilent 1200, Santa Clara, CA, USA). The separations were conducted under gradient conditions using an YMC-Triart C18 column (250 × 4.6 mm, with 5 µm particle size; Kyoto, Japan) (TA12SO5-2546WT) at 40 °C. The mobile phase was water containing 0.5% phosphoric acid and acetonitrile (B), and was run according to the following elution program: 15% B (0–0.5 min), 15–40% B (0.5–28 min), 40–80% B (28–30 min), 80% B (30–31 min), 80–15% B (31–32 min), 15% B (32–40 min). Peaks were detected using an ultraviolet (UV) detector at 360 and 254 nm. The flow rate was 1 mL/min, and the injection volume was 10 µL.

### 4.3. Experimental Animals

Male institute of cancer research (ICR) mice (4–6 weeks old, 20–25 g) were purchased from MJ Co. Ltd. (Seoul, Korea). The mice were divided into several groups comprising 5 per cage with food and water ad libitum in a room maintained at 22 ± 0.5 °C with an alternating 12 h light–dark cycle. The mice were only used once. All animal care and experimental procedures were conducted according to the National Institutes of Health and the ethical guidelines of the International Association for the Study of Pain. The Care and Use of Laboratory Animals were approved by the Animal Care and Use Committee of Hallym University (Registration Number: Hallym 2020-53).

### 4.4. Production of MIA-Induced OA Model

As described by Pitcher’s group, ICR mice were deeply anesthetized until all sensory reflexes ceased. After wiping the right knee joint with 75% ethanol, 10 µL of MIA (100 µg/mL) was injected into the joint space (intra-articular) using Microliter #705 syringes (Hamilton) with 26-gauge needles. Then, the knee was manually extended and bent for 30 s to spread the solution throughout the joint. After 10 days of once-daily intra-articular injection of MIA, the pain threshold was significantly decreased, and this maintained to the 28th day. Oral administration of ME or PC (10, 50, or 100 mg/kg) was initiated 14 days after MIA injection. The von-Frey test was used to assess the pain threshold of mice. According to our previous procedures [52], mice were allowed to adapt to the test environment for 30 min by being placed in a transparent glass cell with a metal mesh floor. The von-Frey filament (North Coast Medical, Inc., Gilroy, CA, USA) was then applied to the plantar surface using the up and down paradigm [53]. Then, the mice hind paw withdrawal threshold was compared.

### 4.5. Plasma Cytokines ELISA Assay in ICR Mice

The plasma cytokines levels, including TNF-α, CRP, and IL-6, were measured at 2 h after oral treatment with the drug. In total, 1 mL of blood was collected by puncturing the retro-orbital venous plexus, followed by centrifugation at 4 °C to separate plasma, which was stored at −80 °C before analysis. The level of CRP in the plasma was evaluated using the Mouse CRP ELISA kit (E-EL-M0053, Elabscience Biotechnology Inc., Houston, TX, USA). The measurement of plasma level of TNF-α was performed using the Mouse TNF-α ELISA kit (MTA00B, R&D Systems, Minneapolis, MN, USA). The level of plasma IL-6 was also examined using the Mouse IL-6 ELISA kit (M6000B, R&D Systems, Minneapolis, MN, USA).

### 4.6. The Measurement of Ear Edema

In total, 30 ICR mice were divided into 6 groups comprising 5 animals each. The mice were treated with different doses of ME or PC (50 mg/kg and 100 mg/kg), and immediately after that, the test was performed. A topical application of 2.5% croton oil (in acetone, 20 µL per ear) was conducted on each mouse’s left ear, as previously described [54]. The animals were euthanized by cervical dislocation 5 h after the croton oil (2.5%) treatment. An 8 mm segment was then removed from each animal’s right ear for weighing in an analytical balance. The weight difference between the injected ear and the untreated ear was compared to determine edema.

### 4.7. Cell Culture and Treatment

RAW 264.7 cells (ATCC^®^ TIB-71^TM^, ATCC, Manassas, USA) were cultured in Dulbecco’s modified Eagle’s medium (DMEM) containing 10% heat-inactivated fetal bovine serum (FBS), 100 U/mL penicillin, and 100 µg/mL streptomycin at 37 °C in a humidified 5% CO_2_ atmosphere, as described [55]. The drugs used for the present study were dissolved in 20% dimethyl sulfoxide (DMSO), and the actual concentration of DMSO in the treated plate was 0.1%.

### 4.8. Measurement of Nitric Oxide

RAW 264.7 cells (1 × 10^5^ cells/well) were seeded in a 96-well plate. After overnight incubation, different concentrations of ME or PC were added 1 h prior to 1 µg/mL LPS for 24 h. The supernatants of culture media were collected after centrifugation for the measurement of nitrite (an indicator of NO production). In total, 100 µL of sample along with 100 µL of Griess reagent (modified, Sigma-Aldrich Co., Louis, MO, USA) was incubated at room temperature for 15 min. We read the absorbance at 540 nm using a Spectramax M2/e fluorescence microplate reader (Molecular Devices). Sodium nitrite was set as the standard, as previously described [56].

### 4.9. Determination of PGE_2_, LTB_4_, and IL-6 Production in the RAW 264.7 Cells

RAW 264.7 cells (1 × 10^5^ cells/well, in a 96-well plate) were treated with different concentrations of ME or PC for 1 h followed by stimulation with LPS (1 µg/mL) for 24 h. Then, supernatants were harvested and assayed for IL-6 (M6000B, R&D Systems) by ELISA kit according to the manufacturer’s instruction. RAW 264.7 cells (1 × 10^6^ cells/well, in a 24-well plate) were stimulated with different concentrations of ME or PC and LPS (1 µg/mL) for 24 h, and the cell culture supernatants were collected. The contents of PGE_2_ (EM1503, Wuhan Fine Biotech Co.,Ltd., Wuhan, China) and LTB_4_ (CSB-E08034m, Cusabio Technology LLC, Wuhan, China) in the supernatants were measured using an ELISA kit according to the manufacturer’s protocol.

### 4.10. Western Blot Analysis

RAW 264.7 cells (1 × 10^6^ cells/well, in a 24-well plate) were pretreated with ME or PC for 1 h and stimulated with LPS (1 µg/mL) for 24 h. Then, the cells were harvested. According to our previous procedure [52], proteins were extracted using the lysis buffer and quantified with the Bradford method (Bio-Rad Laboratories, Hercules, CA, USA). The samples were separated by sodium dodecyl sulfate-polyacrylamide gel electrophoresis (SDS-PAGE), then transferred to polyvinylidene fluoride (PVDF) membranes (Millipore, Bedford, MA, USA). After blocking, the membranes were immunoblotted with primary antibodies TNF-α (0.2 µg/mL, Abcam, Cambridge, UK), p-ERK (1:1000, Cell Signaling Technology, Danvers, MA, USA), p-JNK (1:1000, Cell Signaling Technology, Danvers, MA, USA), p-PPAR-γ (1:1000, Flarebio Biotech LLC, College Park, MD, USA) and β-actin (1:1000, Cell Signaling Technology, Danvers, MA, USA) at 4 °C overnight. After washing, the membranes were incubated for 1 h at 37 °C with the horseradish peroxidase (HRP) conjugated secondary antibodies (1:4000, Enzo Life Sciences, Lausen, Switzerland). Enhanced chemiluminescence reagent (Millipore, Billerica, MA, USA) was added followed by detection of light emission using a Luminescent Image Analyzer (LAS-4000, Fuji Film Co., Tokyo, Japan). The intensity of bands was measured with Multi-Gauge Version 3.1 (Fuji Film Co., Tokyo, Japan). The value of each sample was expressed as the percentage of the control tested protein in the internal reference β-actin.

### 4.11. Statistical Analysis

The data were expressed as the means ± standard deviations (SDs) or standard error of the means (SEMs). GraphPad Prism (Version 8.4.2, GraphPad Software, San Diego, CA, USA) was used to perform the statistical analysis. In Figure 2, the comparison between multiple groups with several time points is assessed by repeated measurement ANOVA, followed by Tukey’s post hoc test. In Figure 3, Figure 4, Figure 6, Figure 7 and Figure 8, one-way ANOVA with Tukey’s post hoc test was used to assess differences between the two groups. In Figure 5, data are compared and analyzed using two-way ANOVA with Tukey’s post hoc test. The unpaired *t*-test assessed differences between the two groups. *p* < 0.05 was statistically significant.

## 5. Conclusions

Our findings suggest that ME has the potential to prevent MIA-induced OA due to its antinociceptive and anti-inflammatory activities. Additionally, the molecular mechanism grounding the antinociceptive and anti-inflammatory effects of ME may involve the inhibition of pro-inflammatory cytokines, the upregulation of anti-inflammatory cytokines, the suppression of inflammatory mediators, and the blocking of the MAPK signaling pathway. Further in vivo studies verified that ME ameliorates the progression of ear edema in mice. Taken together, ME could be considered as a new potential source useful for relieving arthritis.

## Figures and Tables

**Figure 1 plants-10-01234-f001:**
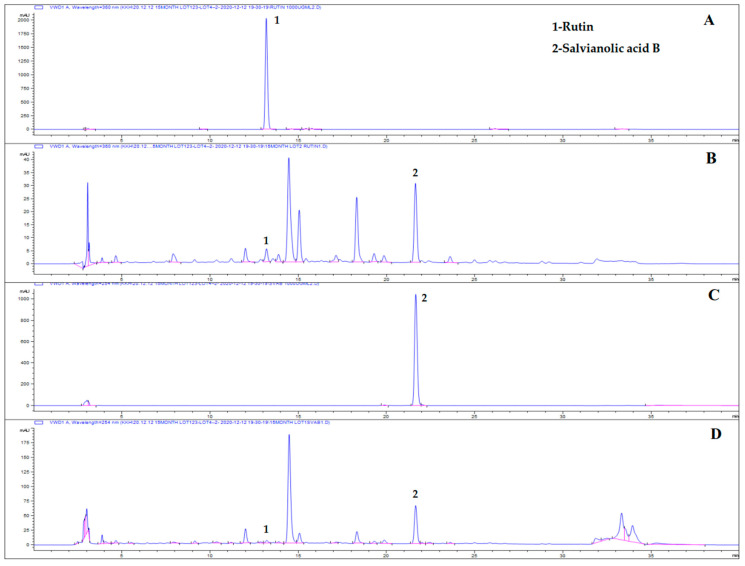
Representative HPLC chromatograms of rutin (**A**) at 360 nm, salvianolic acid B (**C**) at 254 nm, and ME of AP and SM at (**B**) 360 nm and (**D**) 254 nm.

**Figure 2 plants-10-01234-f002:**
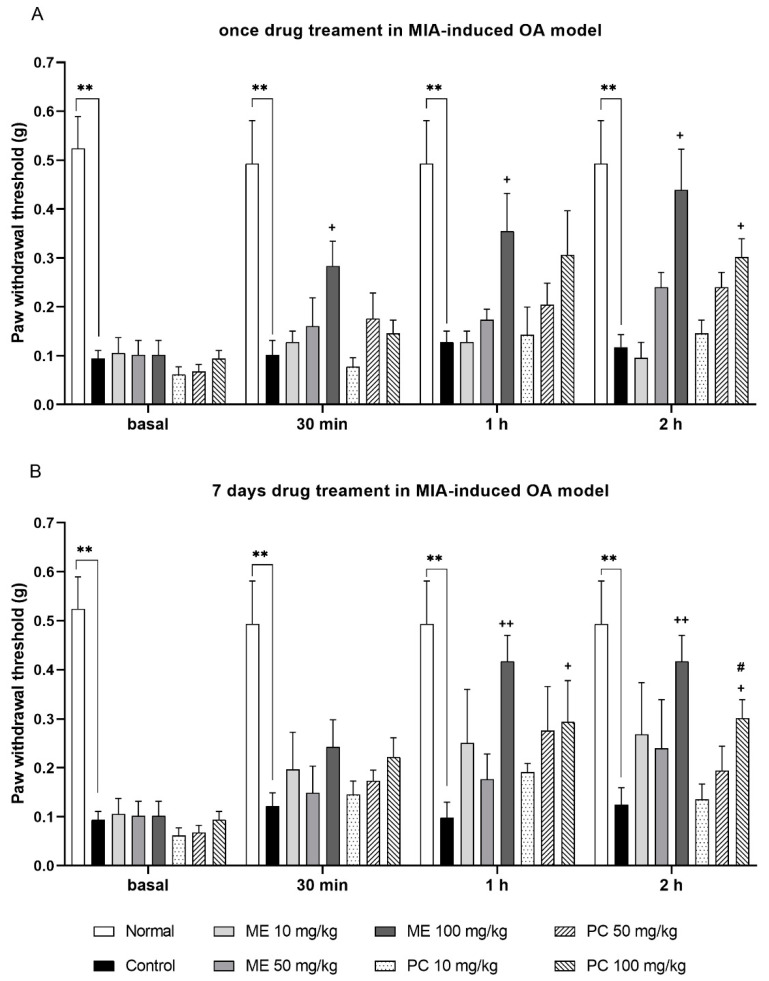
The antinociceptive effect of ME and PC administered orally in the MIA-induced OA model. Mice were administered orally with the vehicle, ME (from 10 to 100 mg/kg), or PC (from 10 to 100 mg/kg), and then were measured for pain threshold at 30, 60, and 120 min after treatment using the von-Frey test. (**A**) Mice were treated with the vehicle, ME, or PC once (basal: Normal vs. Control *p* = 0.0073; 30 min: Normal vs. Control *p* = 0.0089, Control vs. ME 100 mg/kg *p* = 0.0313; 1 h: Normal vs. Control *p* = 0.0095, Control vs. ME 100 mg/kg *p* = 0.0182; 2 h: Normal vs. Control *p* = 0.0084, Control vs. ME 100 mg/kg *p* = 0.0137, Control vs. PC 100 mg/kg *p* = 0.0423). (**B**) Mice were administered orally with the vehicle, ME, or PC for 1 week (basal: Normal vs. Control *p* = 0.0067; 30 min: Normal vs. Control *p* = 0.0099, Control vs. ME 100 mg/kg *p* = 0.0313; 1 h: Normal vs. Control *p* = 0.0089, Control vs. ME 100 mg/kg *p* = 0.0098, Control vs. PC 100 mg/kg *p* = 0.0159; 2 h: Normal vs. Control *p* = 0.0085, Control vs. ME 100 mg/kg *p* = 0.0098, Control vs. PC 100 mg/kg *p* = 0.0276, ME 100 mg/kg vs. PC 100 mg/kg *p* = 0.0463). The values denote the mean ± SEM (*n* = 5). The repeated measurement two-way ANOVA analyzed each quantified result with a Tukey’s post hoc test. ***p* < 0.01; ^+^*p* < 0.05, ^++^*p* < 0.01; ^#^*p* < 0.05.

**Figure 3 plants-10-01234-f003:**
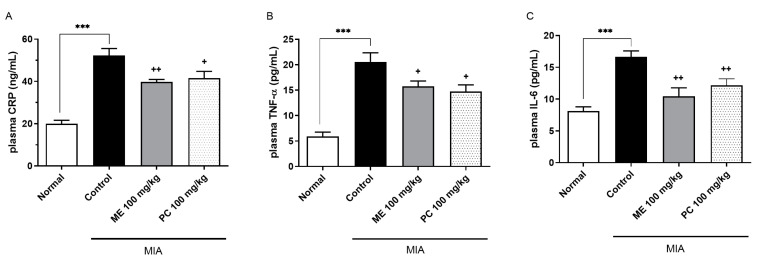
Effect of ME and PC on plasma CRP, TNF-α, and IL-6 levels in the MIA-induced OA model. ME or PC was orally administered to the mice at the dose of 100 mg/kg. The plasma CRP (**A**), TNF-α (**B**), and IL-6 (**C**) levels were measured at 1 h after treatment. The blood was collected from the tail-vein. Error bars represent SEM (*n* = 5). (One-way ANOVA for multiple comparisons, plasma CRP: Normal vs. Control *p* < 0.0001, Control vs. ME 100 mg/kg *p* = 0.0049, Control vs. PC 100 mg/kg *p* = 0.0171; plasma TNF-α: Normal vs. Control *p* < 0.0001, Control vs. ME 100 mg/kg *p* = 0.0340, Control vs. PC 100 mg/kg *p* = 0.0114; plasma IL-6: Normal vs. Control *p* = 0.0002, Control vs. ME 100 mg/kg *p* = 0.0022, Control vs. PC 100 mg/kg *p* = 0.0094). ****p* < 0.001; ^+^*p* < 0.05, ^++^*p* < 0.01.

**Figure 4 plants-10-01234-f004:**
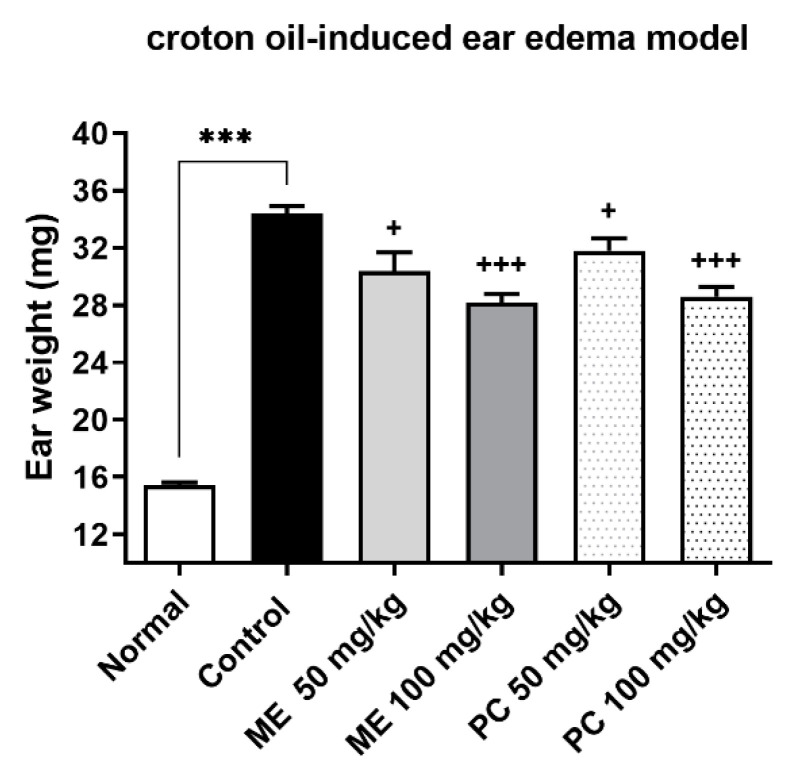
The anti-inflammatory effect of ME and PC administered orally in the ear edema model. ME or PC was orally administered to the mice at 50 and 100 mg/kg. The weight of the ear was measured at 1 h after treatment. Error bars represent SEM (*n* = 5) (one-way ANOVA for multiple comparisons, Normal vs. Control *p* < 0.0001, Control vs. ME 50 mg/kg *p* = 0.0150, Control vs. ME 100 mg/kg *p* < 0.0001, Control vs. PC 50 mg/kg *p* = 0.0329, Control vs. PC 100 mg/kg *p* < 0.0001). ****p* < 0.001; ^+^*p* < 0.05, ^+++^*p* < 0.001.

**Figure 5 plants-10-01234-f005:**
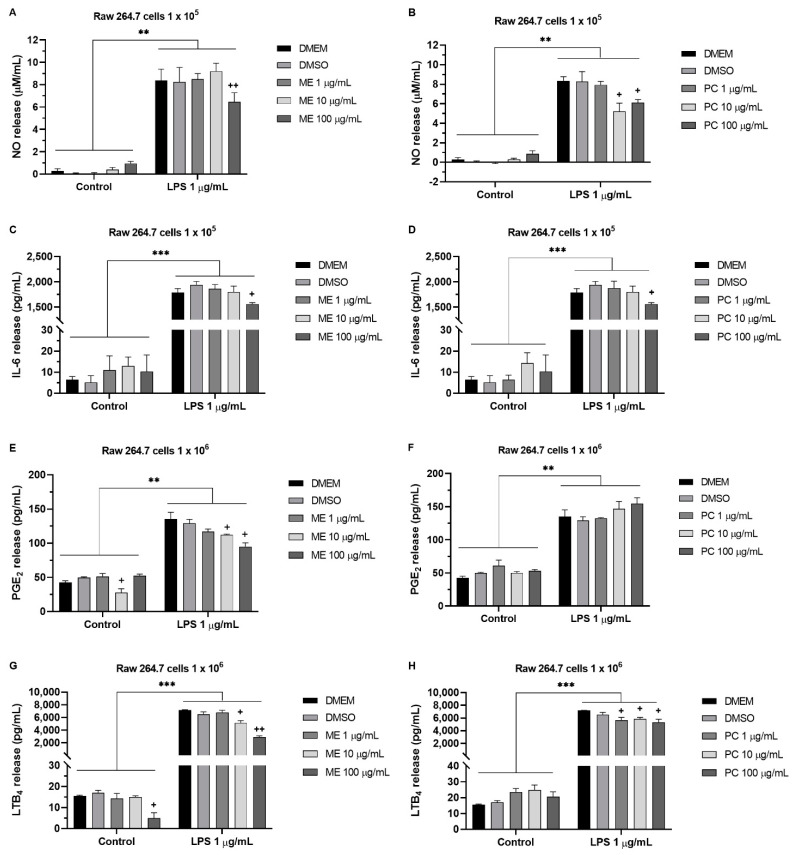
The effect of ME and PC on LPS-induced NO, IL-6, PGE_2_, and LTB_4_ release from cultured RAW 264.7 cells. Several concentrations (1, 10, or 100 µg/mL) of ME or PC were added into the media 1 h prior to LPS treatment. NO (**A**,**B**), IL-6 (**C**,**D**), PGE_2_ (**E**,**F**), and LTB_4_ (**G**,**H**) levels in the medium were measured after RAW cells were incubated with LPS for 24 h. The vertical bars indicate the mean ± SD (*n* = 3). NO release: (**A**) Control vs. LPS 1 µg/mL *p* = 0.0039, DMSO vs. ME 100 µg/mL *p* = 0.0097; (**B**) Control vs. LPS 1 µg/mL *p* = 0.0048, DMSO VS. PC 10 µg/mL *p* = 0.0366, DMSO VS. PC 100 µg/mL *p* = 0.0437. IL-6 release: (**C**) Control vs. LPS 1 µg/mL *p* = 0.0001, DMSO vs. ME 100 µg/mL *p* = 0.0166; (**D**) Control vs. LPS 1 µg/mL *p* < 0.0001, DMSO VS. PC 100 µg/mL *p* = 0.0173. PGE_2_ release: (**E**) Control vs. LPS 1 µg/mL *p* = 0.0024, DMSO vs. ME 10 µg/mL *p* = 0.0498, DMSO vs. ME 100 µg/mL *p* = 0.0361; (**F**) Control vs. LPS 1 µg/mL *p* = 0.0076, DMSO vs. PC 100 µg/mL *p* = 0.0745. LTB_4_ release: (**G**) Control vs. LPS 1 µg/mL *p* < 0.0001, DMSO vs. ME 10 µg/mL *p* = 0.0166, DMSO vs. ME 100 µg/mL *p* = 0.0091; (**H**) Control vs. LPS 1 µg/mL = 0.0001, DMSO vs. PC 1 µg/mL *p* = 0.0151, DMSO vs. PC 10 µg/mL *p* = 0.0412, DMSO vs. PC 100 µg/mL *p* = 0.0293. Two-way ANOVA analyzed each quantified result with a Tukey’s post hoc test. ***p* < 0.01, ****p* < 0.001; ^+^*p* < 0.05, ^++^*p* < 0.01.

**Figure 6 plants-10-01234-f006:**
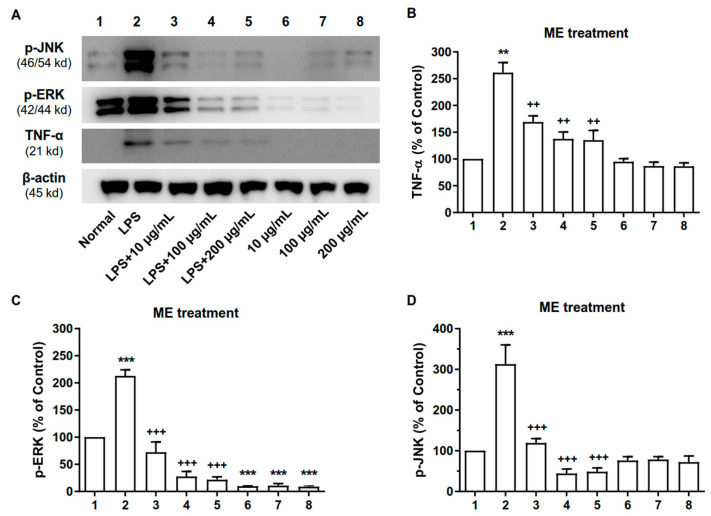
Effect of ME on p-JNK, p-ERK, and TNF-α, proteins in RAW 264.7 cells. (A) The expression levels of p-JNK, p-ERK, and TNF-α proteins were measured by performing Western blotting analysis. The Raw 264.7 cells were treated with ME for 24 h. 1—Normal group, 2—LPS-treated control group, 3—LPS-treated control group + 10 µg/mL ME, 4—LPS-treated control group + 100 µg/mL ME, 5—LPS-treated control group + 200 µg/mL ME, 6—Normal group + 10 µg/mL ME, 7—Normal group + 100 µg/mL ME, 8—Normal group + 200 µg/mL ME. Quantitative analysis of the results of the Western blotting analysis are shown in (**B**) for p-JNK, in (**C**) for p-ERK, and in (**D**) for TNF-α proteins. β-Actin was used as an internal control. Values are mean ± SEM (*n* = 3). These values are expressed as the percentage of the control tested protein /β-actin for each sample. TNF-α (**B**): 1 vs. 2 *p* = 0.0016, 2 vs. 3 *p* = 0.0094, 2 vs. 4 *p* = 0.0077, 2 vs. 5 *p* = 0.0082; p-ERK (**C**): 1 vs. 2 *p* < 0.0001, 2 vs. 3 *p* < 0.0001, 2 vs. 4 *p* < 0.0001, 2 vs. 5 *p* < 0.0001, 1 vs. 6 *p* < 0.0001, 1 vs. 7 *p* < 0.0001, 1 vs. 8 *p* < 0.0001; p-JNK (**D**): 1 vs. 2 *p* = 0.0002, 2 vs. 3 *p* = 0.0004, 2 vs. 4 *p* < 0.0001, 2 vs. 5 *p* < 0.0001. (one-way ANOVA for multiple comparisons, ***p* < 0.01, ****p* < 0.001; ^++^*p* < 0.01, ^+++^*p* < 0.001).

**Figure 7 plants-10-01234-f007:**
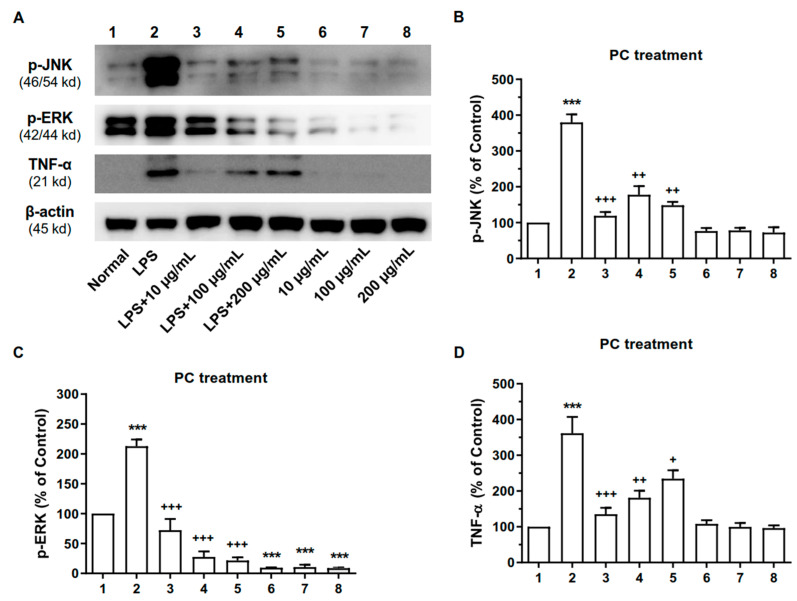
Effect of PC on p-JNK, p-ERK, and TNF-α proteins in RAW 264.7 cells. (**A**) The expression levels of TNF-α, p-ERK, and p-JNK proteins were measured by performing Western blotting analysis. The Raw 264.7 cells were treated with PC for 24 h. 1—Normal group, 2—LPS-treated control group, 3—LPS-treated control group + 10 µg/mL PC, 4—LPS-treated control group + 100 µg/mL PC, 5—LPS-treated control group + 200 µg/mL PC, 6—Normal group + 10 µg/mL PC, 7—Normal group + 100 µg/mL PC, 8—Normal group + 200 µg/mL PC. Quantitative analysis of the results of the Western blotting analysis are shown in (**B**) for p-JNK, in (**C**) for p-ERK, and in (**D**) for TNF-α proteins. β-Actin was used as an internal control. Values are mean ± SEM (*n* = 3). These values are expressed as the percentage of the control tested protein/β-actin for each sample. p-JNK (**B**): 1 vs. 2 *p* < 0.0001, 2 vs. 3 *p* = 0.0001, 2 vs. 4 *p* = 0.0014, 2 vs. 5 *p* < 0.0146; p-ERK (**C**): 1 vs. 2 *p* = 0.0001, 2 vs. 3 *p* = 0.0002, 2 vs. 4 *p* < 0.0001, 2 vs. 5 *p* < 0.0001, 1 vs. 6 *p* < 0.0001, 1 vs. 7 *p* < 0.0001, 1 vs. 8 *p* < 0.0001; TNF-α (**D**): 1 vs. 2 *p* = 0.0006, 2 vs. 3 *p* = 0.0004, 2 vs. 4 *p* = 0.0057, 2 vs. 5 *p* = 0.0066. (one-way ANOVA for multiple comparisons, ***p* < 0.01, ****p* < 0.001; ^+^*p* < 0.05, ^++^*p* < 0.01, ^+++^*p* < 0.001).

**Figure 8 plants-10-01234-f008:**
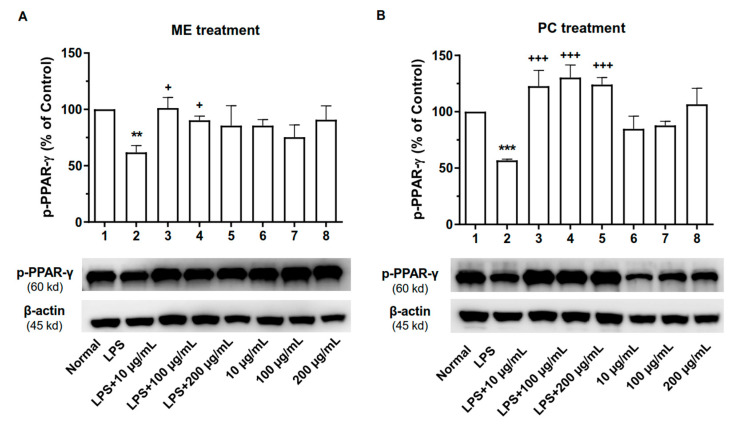
Effect of ME and PC on p-PPAR-γ protein in RAW 264.7 cells. The expression level of p-PPAR-γ protein was measured after ME (**A**) or PC (**B**) treatment in RAW 264.7 cells. 1—Normal group, 2—LPS-treated control group, 3—LPS-treated control group + 10 µg/mL ME or PC, 4—LPS-treated control group + 100 µg/mL ME or PC, 5—LPS-treated control group + 200 µg/mL ME or PC, 6—Normal group + 10 µg/mL ME or PC, 7—Normal group + 100 µg/mL ME or PC, 8—Normal group + 200 µg/mL ME or PC. β-Actin was used as an internal control. Values are mean ± SEM (*n* = 3). These values are expressed as the percentage of the control tested protein/β-actin for each sample. ME-treatment (B): 1 vs. 2 *p* = 0.0091, 2 vs. 3 *p* = 0.0296, 2 vs. 4 *p* = 0.0457; PC-treatment (C): 1 vs. 2 *p* = 0.0009, 2 vs. 3 *p* = 0.0008, 2 vs. 4 *p* = 0.0002, 2 vs. 5 *p* = 0.0004. (one-way ANOVA for multiple comparisons, ***p* < 0.01, ****p* < 0.001; ^+^*p* < 0.05, ^+++^*p* < 0.001).

## Data Availability

All the data used in this study have been provided in the main text and the Appendix A.

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
