# Peer review of "The Anti-Inflammatory and the Antinociceptive Effects of Mixed Agrimonia pilosa Ledeb. and Salvia miltiorrhiza Bunge Extract"

_plants, 2021, doi:10.3390/plants10061234_

Round 1

Reviewer 1 Report

This manuscript described the anti-inflammatory and antinociceptive effects and the mechanism of ME. The study is well designed and interesting. However, there are some questions and comments as below:

  1. In Line 93, it was stated that it had been administered after 10 days, but it was stated that it had been administered after 14 days in the method (line 314-315). Which one is it?
  2. Why did the author choose rutin and Salvianolic acid B in HPLC analysis? Why didn’t the author analyze other functional ingredients? The author only briefly mentioned (line234-236) in the discussion, which is a pity. If the explanation was strengthened, it would be much better.
  3. In Line 215-216, the expression here was not clear. The original explanation was only the difference in wavelength during analysis, and did not represent the difference in composition. It is recommended that the author should explain more clearly.
  4. In line 267-273, the author compared different animal models, and only compared the dose that ME was better than AP or SM alone. This is not appropriate and the evidence seems insufficient. The author should consider the explanation in this section.
  5. In terms of overall layout, it is recommended that the animal experiment part should be put together, such as Line 198 (2.7. ME and PC ameliorated ear edema induced by croton oil.). This section should be put together with the previous animal experiment.

Author Response

Overall comment:  This manuscript described the anti-inflammatory and antinociceptive effects and the mechanism of ME. The study is well designed and interesting. However, there are some questions and comments as below:

Response: Thank you very much for your valuable suggestions, we have revised the manuscript to improve the quality of the current manuscript.

Comment 1: In Line 93, it was stated that it had been administered after 10 days, but it was stated that it had been administered after 14 days in the method (line 314-315). Which one is it?

Response: Thank you very much to point out our mistake. We have corrected this problem by changing ‘10 days’ to ‘14 days.’ Actually, in this experiment, we started oral administration after 2 weeks of the MIA-injected. Please see page 3, line 94.

Comment 2: Why did the author choose rutin and Salvianolic acid B in HPLC analysis? Why didn’t the author analyze other functional ingredients? The author only briefly mentioned (line234-236) in the discussion, which is a pity. If the explanation was strengthened, it would be much better.

Why did the author choose rutin and Salvianolic acid B in HPLC analysis?

Response: Thank you for your consideration. Rutin is one of standing compounds in Agrimonia pilosa Ledeb. and salvianolic acid B is one of the abundant active phytocomponents in Radix  Salvia militiorrhiza root. Then we added some biological activities of these two components in the discussion section. Please see page 10, lines 256-267. In addition, in our previous study (Reference 13), a simple, accurate, and rapid HPLC analysis has been developed to quantify these two polyphenols in mixed extract and was successfully validated. In our present stabilization study, we also selected rutin and Salvianolic acid B as the reference ingredients which could be used as the control of the differences between batches for extraction.

Why didn’t the author analyze other functional ingredients?

Response: The main substituents of Radix  Salvia militiorrhiza dried root are Tanshinone series compounds but those are non-polar compounds, they are little amount resolve in 50% EtOH extraction. And other functional ingredients are not good resolution in HPLC pattern for quantification.

The author only briefly mentioned (line234-236) in the discussion, which is a pity. If the explanation was strengthened, it would be much better.

Response: Thank you for your comment. We added some more explanation in the discussion. Please see page 10, lines 256-267.

Comment 3: In Line 215-216, the expression here was not clear. The original explanation was only the difference in wavelength during analysis, and did not represent the difference in composition. It is recommended that the author should explain more clearly.

Response: In order to avoid confusion, we think we should delete this sentence. Because it is too much confuse for understanding without previous data. We just tried represent the difference in composition of different extraction methods between current method and previous one. In that case, we have to show comparison each data with adding more data from reference 13.

Comment 4: In line 267-273, the author compared different animal models, and only compared the dose that ME was better than AP or SM alone. This is not appropriate and the evidence seems insufficient. The author should consider the explanation in this section.

Response: Thank you for your suggestion. To this issue, we have modified the explanation in the discussion section. Please see page 11, lines 298-306.

Comment 5: In terms of overall layout, it is recommended that the animal experiment part should be put together, such as Line 198 (2.7. ME and PC ameliorated ear edema induced by croton oil.). This section should be put together with the previous animal experiment.

Response: Thank you for your suggestion. We have rearranged the result and method sections. Please see page 5, lines 138-148; page 12, lines 364-372.

Reviewer 2 Report

The present form of the manuscript may be recommended for publication after a minor revision. All authors should read and correct all the minor errors in the main text in the revised manuscript.

Suggestion to clarify Figure 1 for the reader:

  • Mention a number or letter on the top of Rutin peak, for example 1 or a, and Salvianolic 2 orb.
  • Rewrite Figure 1 as follow: Representative HPLC chromatograms of Rutin (A) at 360 nm, Salvianolic acid B (C) at 254 nm, and ME of AP 90 and SM at (B) 360 nm and (D) 254 nm.
  • The authors mentioned the presence of two compounds in the text on page 3 L86-88. It will be more informative If the authors can explain and discuss briefly in the main text the correlation (or difference) of those two mentioned compounds with the current activity results in the mixture based on the previously reported literature. To clarify, L234 to 235 on page 9.

Author Response

Overall comment:  The present form of the manuscript may be recommended for publication after a minor revision. All authors should read and correct all the minor errors in the main text in the revised manuscript.

Comment 1: Suggestion to clarify Figure 1 for the reader: Mention a number or letter on the top of Rutin peak, for example 1 or a, and Salvianolic 2 or b.

Rewrite Figure 1 as follow: Representative HPLC chromatograms of Rutin (A) at 360 nm, Salvianolic acid B (C) at 254 nm, and ME of AP and SM at (B) 360 nm and (D) 254 nm.

Response: Thank you for kindly suggestion. We have revised this sentence. Please see page 3, lines 91-92.

Comment 2: The authors mentioned the presence of two compounds in the text on page 3 L86-88. It will be more informative If the authors can explain and discuss briefly in the main text the correlation (or difference) of those two mentioned compounds with the current activity results in the mixture based on the previously reported literature. To clarify, L234 to 235 on page 9.

Response: Thank you for your consideration. Rutin is one of standing compounds in Agrimonia pilosa Ledeb. and salvianolic acid B is one of the abundant active phytocomponents in Radix  Salvia militiorrhiza root. Then we added some biological activities of these two components in the discussion section. Please see page 10, lines 256-267.

Reviewer 3 Report

Throughout the manuscript: the acronyms has to be reported in extenso at their first appearance in the text.

The ANOVA P values have to be included in the figure captions.

More details about the detection and quantification of the phytochemicals should be included in the material and methods. Did the authors confirm the presence of such phytochemicals with MS analysis?

In figure 1, the phytochemicals of interest should be marked.

Authors should explain the statistical approach followed for the calculation of the number of animals used in the experimental procedures.

Author Response

Comment 1: Throughout the manuscript: the acronyms has to be reported in extenso at their first appearance in the text.

Response: Thank you for your suggestion. We have added the extension word of some acronyms. Please see page 2, lines 64, 86, and 87; page 3, line 110; page 5, line 149; page 6, line 155,

Comment 2: The ANOVA P values have to be included in the figure captions.

Response: Thank you. As you suggested, we have added all the p values in the figure captions. Please see page 4, lines 118-128; page 5, lines 133-137 and 145-147; page 7, lines 186-196; page 8, lines 204-208; page 9, lines 216-220; page 10, lines 228-231.

Comment 3: More details about the detection and quantification of the phytochemicals should be included in the material and methods. Did the authors confirm the presence of such phytochemicals with MS analysis?

Response: Thank you for your consideration. We have described more details about the detection and quantification of the phytochemicals in the material and methods with reference (page 11, lines 314-316). In those references, our results suggested that the HPLC separation combined with an adequate efficiency described in this work allows the application of the method for routine and QA/QC analysis from mixed and complex extracts. And, we confirmed the structure of rutin in Agrimonia pilosa by various spectroscophic techniques including MS analysis in previous works [reference 36]. Salvianolic acid B is very popular & standing compound in Salvia miltiorrhiza and we confirmed by UV pattern & retention time with reference compound.

Comment 4: In figure 1, the phytochemicals of interest should be marked.

Response: Thank you for your comments. According to your comment, rutin and salvianolic acid B which were the phytochemicals of interest marked in the HPLC patterns of  Figure 1, please see page 3, line 90.

Comment 5: Authors should explain the statistical approach followed for the calculation of the number of animals used in the experimental procedures.

Response: Thank you for your kind suggestion. We have added some details of the statistical approach both in the methods section and the figure captions. Please see page 4, lines 126-128; page 5, lines 133 and 145; page 7, lines 186; page 8, lines 203; page 9, lines 215; page 10, lines 227; page 13, lines 420-421.